# Synthesis and Properties of a Bay-Annulated-Indigo Tetramer Based on Low-Cost Spiro[Fluorene-9,9′-Xanthene] Core

**DOI:** 10.3390/molecules24193623

**Published:** 2019-10-08

**Authors:** Bao-Yi Ren, Yan-Mei Tan, Chang-Liang Sun, Yong-Gang Sheng

**Affiliations:** 1College of Applied Chemistry, Shenyang University of Chemical Technology, Shenyang 110142, China; m15124190633@163.com (Y.-M.T.); chemscl@126.com (C.-L.S.); gara@syuct.edu.cn (Y.-G.S.); 2Key Laboratory for Organic Electronics and Information Displays & Institute of Advanced Materials (IAM), Nanjing University of Posts and Telecommunications, 9 Wenyuan Road, Nanjing 210023, China

**Keywords:** bay-annulated indigo, spiro[fluorene-9,9′-xanthene], synthesis, properties

## Abstract

A three-dimensional bay-annulated-indigo (BAI) tetramer has been prepared by appending BAI units onto a low-cost spiro[fluorene-9,9′-xanthene] (SFX) core. The target compound 4BAI-SFX exhibits strong and broad absorption in the visible region covering the range of 450~700 nm. The electrochemical measurement illuminates the characteristics of a deep lowest unoccupied molecular orbital (LUMO) level and multiple redox states of 4BAI-SFX. These results suggest that 4BAI-SFX should be a selectable electron-transporting material for eco-friendly organic semiconductors.

## 1. Introduction

Currently, organic semiconductors are dynamic areas which are attracting intensive interest from both academia and industry due to their rosy prospects for next-generation display, lighting and photovoltaic applications [1,2]. To develop novel building blocks, design high-performance molecules and reduce synthetic costs, researchers have made enormous efforts and achieved remarkable successes. For instance, non-fullerene electron-transporting [3,4] and low-cost hole-transporting materials [5,6] have been rapidly emerging in the last five years for highly efficient photoelectric devices. It is universal and effective pathway for manipulating the electronic structures of organic molecular semiconductors that precisely assemble *p*-type and *n*-type units: the so-called donor-acceptor (D-A) structures. However, relative to the abundant *p*-type units, much less attention has been paid to exploiting new *n*-type building-blocks [7].

Indigo, widely known as an ancient dye, has been used for thousands of years and is produced in amounts of thousands of tons per year. There is a fascinating prospect of improving the processability and electron-transport property of indigo by chemical modification and then transforming it into a novel and inexpensive building block for organic electronics. Inspired by the Cibalackrot, an indigo derivative created by Ciba Specialty Chemicals, Liu and co-workers altered the phenyl groups with thiophene groups; bay-annulated indigo (**BAI**), as a new form of the old natural dye, was first synthesized with the characteristics of two-dimensional conjugation and electron deficiency [8,9]. Using BAIs as versatile electron accepting units, a wide variety of high-performance polymers were designed and prepared for organic electronics and photovoltaic applications [10,11,12,13,14]. Nevertheless, BAI-based small molecules have only been explored in terms of side-chains and conjugated structures [15,16].

For planar fused-ring units, arranging them onto three-dimensional (3D) cores has become a successful strategy with the advantages of weak intermolecular aggregation, morphology stability, and homogeneous isotropic charge transport and optical properties [17]. Spiro[fluorene-9,9′-xanthene] (**SFX**) is a new class of 3D aromatic compound which has drawn great interest recently as a core backbone for “eco-friendly green organic semiconductors” due to its merits of “one-pot” preparation and the binary conjugation of fluorene moiety and xanthene moiety [18,19]. In our previous work, two BAI-dimers (**2,7-BAI-SFX** and **2′,7′-BAI-SFX**), appending **BAI** units on the 2,7- and 2′,7′-positions of **SFX**, were prepared and studied [20]. Herein, we would like to present the design, synthesis and properties of a **BAI** tetramer (denoted as **4BAI-SFX**; see Scheme 1 for structure) constructed on the **SFX** core; the results of photophysical, electrochemistry, and computational study imply that **4BAI-SFX** possesses great potential as a low-cost and *n*-type 3D organic semiconductor.

## 2. Results and Discussion

### 2.1. Synthesis and Cost Analysis

The synthetic route for the preparation of **4BAI-SFX** is shown in Scheme 1. The 3D core of 2,3′,6′,7-tetrabromo-spiro(fluorene-9,9′-xanthene), named **4Br-SFX**, was smoothly synthesized utilizing 2,7-dibromo-9H-fluoren-9-one and 3-bromophenol as starting materials through a full-fledged one-pot, non-solvent procedure [18]. Compared with the classical spirobifluorene (**SBF**), the synthetic cost of **4Br-SFX** is only about one-ninth as much as brominated **SBF** (3.2 $/g for **4Br-SFX**; see Appendix A, see Appendix A). Furthermore, the tetrabromo-derivatives of **SFX** are more diverse in substituted positions than monotonous 2,2′,7,7′-tetrabromo-SBF; that is, the 2,2′,7,7′- and 2,3′,6′,7-positions of **SFX** can all be functionalized expediently by bromine atoms. Then, **4Br-SFX** was converted to **4Bpin-SFX** by Miyaura borylation. Using bromo-BAI [8] as the coupling partner with 4Bpin-SFX, the target compound **4BAI-SFX** was successfully prepared via the Suzuki–Miyaura reaction. In view of the low cost of indigo (several US dollars per kilogram), we also estimated the synthetic cost of bromo-BAI to be about 5.49 $/g (see Appendix A). Therefore, the **4BAI-SFX** can be considered as an inexpensive 3D organic semiconductor.

The structure of **4BAI-SFX** was characterized by ^1^H NMR and MALDI-TOF MS (see Appendix A). In the ^1^H NMR spectrum of **4BAI-SFX** (Appendix A), the double peak at a 4.03 chemical shift was assigned to the eight protons of methylene adjoined with the oxygen atom of 2-ethylhexyloxy side-chains, indicating the successful coupling between **BAI** and **SFX**. **4BAI-SFX** exhibits good solubility in common organic solvents such as chloroform, chlorobenzene and dichlorobenzene, presumably attributed to the 3D molecular geometry and the flexible 2-ethylhexyl chains. The decomposition temperature *T*_d_ of **4BAI-SFX** is measured to be about 389 °C (see the TGA curves in Appendix A), indicating that the compound possesses excellent thermal stability.

### 2.2. Photophysical Properties

The absorption spectra of **4BAI-SFX** in dilute chloroform solution and spin-coated film are shown in Figure 1. Both of the spectra exhibit two groups of featureless absorption bands. In the short-wavelength region (below 400 nm), the absorption bands can be ascribed to the n→π* and π→π* transitions of the **SFX** core; long-wavelength absorptions (above 450 nm) can be attributed to the intramolecular charge-transfer (ICT) transitions of acceptor–donor–acceptor (A-D-A) structures [21], which are BAI–fluorene–BAI and BAI–xanthene–BAI arrangements. The maximum absorption peaks are located at 596 nm for the solution with a large molar absorption coefficient of 1.5 × 10^5^ M^−1^ cm^−1^ and at 593 nm for the film, respectively. The almost identical peak positions of maximum absorptions imply weakened intermolecular interactions of **4BAI-SFX** benefiting from the 3D molecular geometry [3,20]. According to the absorption edge of **4BAI-SFX** in CHCl_3_ solution, the band gap was calculated to be 1.80 eV.

### 2.3. Electrochemical and Computational Studies

To investigate the electrochemical property of **4BAI-SFX**, cyclic voltammetry (CV) was performed using a conventional three-electrode setup and ferrocene as an internal potential standard (Figure 2). It can be noted that **4BAI-SFX** exhibits two quasi-reversible oxidation peaks and reduction peaks in the range of 1.22~1.68 V. The cathodic peak potentials (*E*_pc_) of the reduction peaks are −1.35 and −1.68 V, and the anodic peak potentials (*E*_pa_) of the oxidation peaks are 0.91 and 1.22 V. The existence of multiple redox states of **4BAI-SFX** would be favorable to charge carrier transporting when utilized in organic electronic devices. According to the onset oxidation and reduction potentials, we calculate that the highest occupied molecular orbital (HOMO)and LUMO levels of **4BAI-SFX** are −5.34 and −3.59 eV, respectively. Hence, the bandgap determined by electrochemical measurement is 1.75 eV, which is very close to the optical bandgap (1.80 eV). The properties of the lowered LUMO level and multiple redox states imply that **4BAI-SFX** should be an appropriate semiconductor material for electron acceptance and transport.

To understand the frontier molecular orbital (FMO) distribution of **4BAI-SFX**, the 3D geometries were calculated at the B3LYP/6-31G* level, as shown in Figure 3. The orbital distributions suggest that the HOMO is assigned on the A-D-A conjugated segment, and the LUMO is mainly distributed on the π orbital of BAI units. It indicates that the combination of planar and 3D structures would be a rational strategy to exert the respective advantages of **BAI** and **SFX** units simultaneously. The full results of the photophysics, electrochemistry, and calculations based on density function theory are summarized in Appendix A.

## 3. Materials and Methods

### 3.1. General Information

The solvents were collected from an activated alumina column purification system, and all starting materials were used as received from commercial sources. The ^1^H NMR spectra were recorded on a Bruker Avance 500 II. Chemical shifts (*δ*) of the signals were expressed in ppm relative to the locked deuterated solvent using tetramethylsilane (TMS) as an internal standard, and coupling constants (*J*) were given in Hertz (Hz). The matrix-assisted laser desorption ionization (MALDI-TOF) mass spectra were measured on a Bruker Autoflex III RF200-CID instrument. Thermogravimetric analysis (TGA) was undertaken with a Shimadzu thermogravimeter at a heating rate of 10 °C/min under N_2_. UV-vis absorption spectra were measured on a Cary 5000 UV-Vis-NIR spectrometer in a quartz cell. Cyclic voltammetry (CV) was performed using a LK98B II Microcomputer-based Electrochemical Analyzer, wherein glassy carbon, platinum and a silver wire act as the working electrode, the counter electrode and the pseudo-reference electrode, respectively. Samples were prepared in CHCl_3_ solution with tetrabutylammonium hexafluorophosphate (NBu_4_PF_6_, 0.1 M) as the electrolyte at a scan rate of 100 mV s^−1^, using a ferrocene/ferrocenium (Fc/Fc+) redox couple as an internal standard. The HOMO and LUMO energy levels (eV) of the two compounds are calculated according to the formula − [4.8 eV + E_ox/red_ (vs. E_Fc/Fc+_)]. All the density functional theory (DFT) calculations were performed using the Gaussian 09 suite of programs at the B3LYP/6-31G* level.

### 3.2. Synthetic Procedure

The starting material **4Br-SFX** was synthesized according to the procedure in the literature [18]. The crude product was dried and purified by column chromatography (petroleum ether/ethyl acetate, 6:1) to obtain a white solid (1.91 g, yield = 66%). ^1^H NMR (500 MHz, CDCl_3_) *δ* 7.63 (d, *J* = 8.1 Hz, 2H), 7.52 (d, *J* = 8.1 Hz, 2H), 7.42 (s, 2H), 7.21 (s, 2H), 6.97 (d, *J* = 8.4 Hz, 2H), 6.23 (d, *J* = 8.4 Hz, 2H).

Regarding 2,2′,2″,2″′-(spiro[fluorene-9,9′-xanthene]-2,3′,6′,7-tetrayl)tetrakis(4,4,5,5-tetramethyl-1,3,2-dioxaborolane) (**4Bpin-SFX**), **4Br-SFX** (1.00 g, 1.54 mmol), bis(pinacolato)diboron (2.35 g, 9.24 mmol, 1,1′-bis (diphenylphosphino)ferrocene)dichloro-palladium(II) (230.00 mg, 0.31 mmol), potassium acetate (1.81 g, 18.48 mmol) and 1,4-dioxane 20 mL were placed in a flask and stirred for 24 h under a nitrogen atmosphere at 90 °C. After the completion of heating, the mixture was cooled to room temperature, water was added and transferred to a separating funnel, and the mixture was extracted with chloroform. The organic layer was concentrated under reduced pressure and the concentrate was purified by silica gel column chromatography (petroleum ether/ethyl acetate, 10:1) and then recrystallized from ethanol. A total of 0.61 g of **4Bpin-SFX** was obtained (yield = 47%). ^1^H NMR (500 MHz, CDCl_3_) *δ* 7.82 (t, *J* = 1.1 Hz, 4H), 7.65 (d, *J* = 1.2 Hz, 2H), 7.50 (t, *J* = 0.9 Hz, 2H), 7.14 (dd, *J* = 7.7, 1.2 Hz, 2H), 6.32 (d, *J* = 7.7 Hz, 2H), 1.32 (s, 24H), 1.26 (d, *J* = 6.0 Hz, 24H).

**Bromo-BAI** (7-(5-bromothiophen-2-yl)-14-(4-((2-ethylhexyl)oxy)phenyl)-7,7a,14,14a-tetrahydrodiindolo[3,2,1-de:3′,2′,1′-ij][1,5]naphthyridine-6,13-dione) was synthesized according to previous reporting [8] and was obtained as a purple solid with an overall yield of 52% (0.15 g). ^1^H NMR (500 MHz, CDCl_3_) *δ* 8.52 (dd, *J* = 14.1, 8.1 Hz, 2H), 8.21 (d, *J* = 8.0 Hz, 1H), 7.68 (dd, *J* = 17.9, 8.2 Hz, 3H), 7.62–7.51 (m, 4H), 7.32 (t, *J* = 7.8 Hz, 1H), 7.28–7.20 (m, 1H), 7.10 (d, *J* = 8.7 Hz, 2H), 3.97 (dd, *J* = 5.7, 2.0 Hz, 2H), 1.80 (hept, *J* = 6.3 Hz, 1H), 1.61–1.20 (m, 8H), 0.98 (t, *J* = 7.5 Hz, 3H), 0.94 (dd, *J* = 33.7, 7.0 Hz, 1H).

Regarding 14,14′-(spiro[fluorene-9,9′-xanthene]-2,7-diylbis(thiophene-5,2-diyl))bis(7-(4-((2-ethylhexyl)oxy)phenyl)diindolo[3,2,1-de:3′,2′,1′-ij][1,5]naphthyridine-6,13-dione) (**4BAI-SFX**), Bromo-BAI (170 mg, 0.24 mmol), 4Bpin-SFX (45 mg, 53.5 µmol), palladium tetrakistriphenylphosophine (10 mg, 9.0 µmol) and solvent (toluene/THF, 1:1, 4 mL) were added to a round-bottomed flask with a stir bar. After adding the solution of K_2_CO_3_/KF (2 M, 0.4 mL), the solution was degassed by bubbling argon while stirring for 10 min. Then, the flask was placed in a 90 °C oil bath overnight. Upon completion by TLC, the product was purified by column chromatography (CHCl_3_ eluent) to provide a dark-blue solid of 67 mg; 46% yield; m.p. decomposed at 275 °C in air; ^1^H NMR (500 MHz, C_2_D_2_Cl_4_, 358 K) *δ* 8.56 (t, *J* = 8.6 Hz, 8H), 8.29 (d, *J* = 8.2 Hz, 4H), 8.00 (d, *J* = 9.2 Hz, 2H), 7.95 (d, *J* = 6.3 Hz, 2H), 7.83 (s, 2H), 7.79–7.75 (m, 4H), 7.69 (t, *J* = 8.1 Hz, 12H), 7.63–7.53 (m, 12H), 7.39–7.32 (m, 6H), 7.30–7.22 (m, 6H), 7.14 (dd, *J* = 8.6, 5.2 Hz, 8H), 6.77–6.73 (m, 2H), 4.03 (d, *J* = 5.6 Hz, 8H), 1.88–1.82 (m, 4H), 1.37 (d, *J* = 57.7 Hz, 32H), 1.08–0.95 (m, 24H). HRMS (MALDI-TOF) for C_177_H_136_N_8_O_13_S_4_ ([M]^+^): 2709.9143, found: 2709.9170.

## 4. Conclusions

In summary, the BAI-based tetramer **4BAI-SFX**, appending **BAI** units onto the 2,3′,6′,7-positions of the **SFX** core, has been successfully synthesized using a low-cost route. The photophysical and electrochemistry investigations as well as theoretical calculation indicate that the tetramer exhibits strong absorption in the visible region and *n*-type characteristics with multiple redox states. All these features suggest **4BAI-SFX** to be a promising and eco-friendly electron-transport candidate for applications in organic electronics.

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
