# Peer review of "Synthesis and Properties of a Bay-Annulated-Indigo Tetramer Based on Low-Cost Spiro[Fluorene-9,9′-Xanthene] Core"

_molecules, 2019, doi:10.3390/molecules24193623_

Round 1

Reviewer 1 Report

This manuscript reports on the synthesis of a new molecular semiconductor based on an acceptor-donor-acceptor architecture. The building blocks (i.e., a xanthene-fluorene spiro compound and an annulated indigo derivative) are selected for their well-established potential as donor and acceptor units, respectively. The originality of the work stands in the synthesis of a four-armed structure around the central spiro unit. The electronic properties of the compounds are measured and appear quite promising; a cost analysis is also provided, which is welcome to assess the economic potential of the compound. This work is clearly of strong interest for the community of organic electronics and should be published in Molecules. Only a couple of technical glitches should be corrected:

The value of the molar absorption coefficient (section 3.2) should read ‘1.5 x 10 to the power of 5’ instead of ‘1.5 x 105’. The HOMO and LUMO energies deduced from the electrochemical measurements are different in the text of section 3.3 (-5.34 and -3.59 eV, respectively) and in Table S3 (-5.67 and -3.92 eV, respectively).

Author Response

Thanks a lot for your genial and careful comments! All of the technical glitches had been corrected. In Table S3, the HOMO and LUMO energies were revised to -5.34 and -3.59 eV, respectively.

Reviewer 2 Report

The authors present their work of the synthesis of a organic semiconductor compound (BAI-SFX). The data could support the conclusion. It could be a very good paper if the following data could be collected.

Include an optical image of the cast film (spin coating or drop casting) under UV light. Please also collect the photoluminescence spectra. If possible, please fabricate a OLED device or LEC device using the BAI-SFX with current-voltage-luminance measurement. Some device fabrication information could be found by literature published by Franky So from NC State Univerisity, and Qibing Pei from UCLA.

Author Response

Response:

Your comments are constructive, Thanks!

For organic semiconductors, it is an important principle that a good OPV should be a good OLED too. Indeed, we observed the basic phenomenon of photoluminescence. An optical image of 4BAI-SFX solution under UV light was provided as below, and a yellow-light compound (synthesized in our lab with a measured PLQY of 71%) was showed together for comparison. The emitting of 4BAI-SFX is very weak, so the photoluminescence spectra and devices fabrication did not be conducted.  

For illuminating the assignments of UV-vis spectrum, a paper of Franky So was added as Ref. 21.

“21. Tsang, S. W.; Chen, S; So, F. Energy level alignment and sub‐bandgap charge generation in polymer: fullerene bulk heterojunction solar cells. Adv. Mater. 2013, 25, 2434-2439.”

Reviewer 3 Report

The bay-annulated-indigo (BAI) is an interesting electron accepting molecule with potential applications in organic electronics. On the other hand, the SFX derivatives can be applied as good hole transporting materials, e.g. for organic light-emitting diodes. The Authors combined these two promising for application molecules and synthesized a new molecule 4BAI-SFX  with three-dimensional structure (3D). Without doubt this is a remarkable compound, most probably with 3D semiconducting properties and potentially interesting for molecular electronics. However, in my opinion it is an abuse to discuss about its semiconducting properties without electrical conductivity measurements of bulk samples or thin films. I suggest that either the electrical measurements should be done or the discussion and conclusions should be modified (at this level of research it is not sure that 4BAI-SFX is a 3D semiconductor and a good candidate for applications in organic electronics). Moreover, I have the following remarks:

Line 125. Here we see a remark about NMR and MALDI-TOF spectra shown in Figs. S1-S5 in Supplementary Materials without any discussion. I think some most important conclusions resulting from these spectra should be given. Lines 135-136. I think that the assignment of absorption bands it is reasonable, nevertheless arguments (or suitable citations), supporting the proposes assignment, are necessary. Lines 141-140. I do not agree that “… almost identical peak positions of maximum absorptions …” give evidence of “… weakened intermolecular interactions …”. It would be strange if these spectra were different. On the other hand, the band of film at 593 nm is broader in comparison with that one of solution, what is most probably  due to intermolecular interactions. I think that more detailed description of DFT calculations should be given.

Author Response

1. 

Thanks! the discuss about the NMR spectrum of 4BAI-SFX was given as below.

“In the 1H NMR spectrum of 4BAI-SFX (Figure S-4), the double peak at 4.03 chemical shift  was assigned the eight protons of methylenes adjoined with the oxygen atom of 2-ethylhexyloxy side-chains, indicating the successful coupling between BAI and SFX.”

2. 

OK! A new citation was added as ref. 21.

“21. Tsang, S. W.; Chen, S; So, F. Energy level alignment and sub‐bandgap charge generation in polymer: fullerene bulk heterojunction solar cells. Adv. Mater. 2013, 25, 2434-2439.”

3. Thanks! If there are noteworthy H-aggregation or J-aggregation between molecules, the maximum absorption peak of corresponding UV-vis spectra would show remarkable blue-shift or red-shift. Hence, we give the our opinion:implying weakened intermolecular interactions. This is also a preliminary result.
